# Are Aquavoltaics Investable? A Framework for Economic and Environmental Cost-Benefit Analysis

**Lihchyi Wen** [1]**, Chun-Hsu Lin** [2,*] 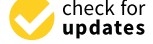 **and Ying-Chiao Lee** [2]

1    ERM Taiwan Co., Ltd., Taipei 104, Taiwan; lihchyi.wen@erm.com
2    Chung-Hua Institution for Economic Research, Taipei 106, Taiwan; yclee@cier.edu.tw
*    Correspondence: chlin@cier.edu.tw; Tel.: +886-2-273-560-06 (ext. 509)

**Abstract:** Aquaculture has long been a significant industry in Taiwan, contributing significantly to the country's GDP through both exports and domestic consumption. However, certain aquaculture practices have faced criticism due to their heavy groundwater usage, resulting in environmental damage such as land subsidence in the southwestern region of Taiwan. In order to change the industry's negative environmental image and achieve the ambitious targets set by the Taiwanese government, including 20 GW of solar photovoltaic power by 2025 and net-zero carbon emissions by 2050, the utilization of aquaculture lands, particularly aquaculture ponds, has emerged as a promising option for solar power development. As the government promotes the symbiosis of aquaculture and solar PV power to attain its renewable energy goals, various stakeholders have engaged in discussions surrounding this approach. Consequently, it is crucial to assess the costs and benefits of such integrated practices from both economic and environmental perspectives, as it will play a pivotal role in shaping the future of the industry. A comparative analysis reveals that an aquaculture–electricity symbiosis with a capacity of 227 MW can further reduce carbon emissions by approximately 150,393.6 tons of CO2e per year, along with reductions of 56.8 tons/year of SOx, 82.3 tons/year of NOx, 3.7 tons/year of PM2.5, and 4.6 tons/year of PM10. These environmental benefits are equivalent to approximately TWD 7626.43 million annually. (Note: CO2e refers to carbon dioxide equivalent, SOx refers to sulfur oxides, NOx refers to nitrogen oxides, PM2.5 refers to fine particulate matter, and PM10 refers to particulate matter with a diameter of 10 μm or less).

**Keywords:** aquavoltaics; system dynamics models; cost-benefit analysis; environmental cost

## 1. Introduction

Before the outbreak of COVID-19 in early 2020, many major countries and economies had embraced "Green New Deals" as a crucial economic development strategy. The pandemic, one of the most severe in human history, further emphasized the need for green development in the post-COVID recovery era. A key focus of these Green New Deals implemented by various countries has been the pursuit of carbon neutrality, which aligns with the broader sustainability agenda. Among the diverse range of green solutions, renewable energy has emerged as a top choice for climate policies worldwide. Solar photovoltaics, in particular, have gained significant popularity among the renewable energy options. According to annual assessments and statistics provided by esteemed international think tanks, such as Bloomberg New Energy Finance (BNEF) and the International Renewable Energy Agency (IRENA), renewable energy has become a dominant force, accounting for 80% of new energy installation capacity globally [1,2]. In 2019, the global capacity of newly installed renewable energy reached 176 GW, with solar photovoltaics constituting 118 GW (67%). In 2020, the global capacity climbed further to 260 GW, with solar photovoltaics contributing 127 GW (49%). Due to its significantly lower environmental impact compared to coal-fired power generation, solar power is recognized as a crucial technology for carbon reduction [3]. BNEF predicts that, by 2050, approximately 62% of the energy in the global

power system will be derived from renewable sources, with solar photovoltaics being one of the most prominent renewable energy sources.

In the pursuit of carbon neutrality, or net-zero goals, within the aquaculture industry, which heavily relies on electricity and water, the symbiosis of fish farming and solar electricity has gained traction in Asia and Nordic countries where aquaculture is prevalent such as China, Taiwan, Indonesia, Malaysia, Canada, Bangladesh, and Vietnam [4]. In particular, the symbiotic practice of shrimp farming not only provides favorable conditions for roof-type solar PV power development but also improves the working environment for shrimp farm employees by providing shade and maintaining a constant water temperature. This approach also helps prevent external predation and disease outbreaks, leading to a substantial increase in shrimp or fish yields. Furthermore, compared to traditional shrimp farming, a 1 MW shrimp–electricity symbiosis system can reduce approximately 15,000 tons of CO2e emissions per year from renewable power generation and decrease water consumption by 75% due to reduced evaporation and cooling requirements. Additionally, the development of fish farming and electricity symbiosis for indoor aquaculture has the potential to significantly reduce the land area needed for photovoltaic installations.

## 2. Materials and Methods

The experiences with aquaculture photovoltaics demonstrate that, in addition to reducing carbon emissions, they can effectively safeguard aquaculture and potentially serve as a model for food, drinking water, and energy production in the face of potential climate change catastrophes [5,6]. The symbiosis of aquaculture and electricity generation aligns with Taiwan's 2025 national green energy goal and is considered a crucial approach. However, socially, there have been media reports highlighting disputes related to the hasty conversion of farmland for electricity generation, infringement on the rights and interests of tenant farmers, reductions in aquacultural production, and improper ownership transfers. In the photovoltaic operation industry, risks such as inadequate grid infrastructure, complex administrative processes, policy changes, local resistance, and natural disasters also exist. Moreover, with the increasing diversity of domestic solar photovoltaic installations, standardized management is crucial, particularly regarding material selection and construction quality control. Consequently, it is essential to clarify the cost structure of "Aquavoltaics" and identify the key factors influencing solar photovoltaic power generation goals and public perception. Currently, there is a lack of comprehensive cost-benefit assessments concerning the lifespan of aquaculture–electricity symbiosis, leading to disputes and conflicting interests. Insufficient attention has been given to whether the symbiosis model is suitable for renewable energy development.

To facilitate the integration of information and provide clear assessments, this study aims to foster rational dialogue among the general public, the private sector, and the media. Through a cost-benefit analysis perspective, the research team has collected, consolidated, and analyzed significant aquaculture photovoltaic projects. Additionally, the study incorporates system dynamics modeling (SDM) to assess the cost-effectiveness of aquavoltaics over their operational lifespans. Figure 1 illustrates the overall process of SDM modeling for this project.

To assess the costs and benefits of photovoltaic development in aquaculture under different scenarios, we initially collected financial data through interviews from representative cases of aquaculture and PV–electricity symbiosis. Concurrently, official data released by the Council for Agriculture were compiled in a system dynamics model (SDM) to evaluate the additional environmental benefits compared to pure aquaculture without symbiosis over a 20-year operational period. Operational information was gathered to evaluate the economic cost-effectiveness of a 227 MW ground-based aquaculture–photovoltaic system using financial analysis. The financial analysis was conducted from the perspectives of aquaculture farmers, photovoltaic system operators, and society as a whole.

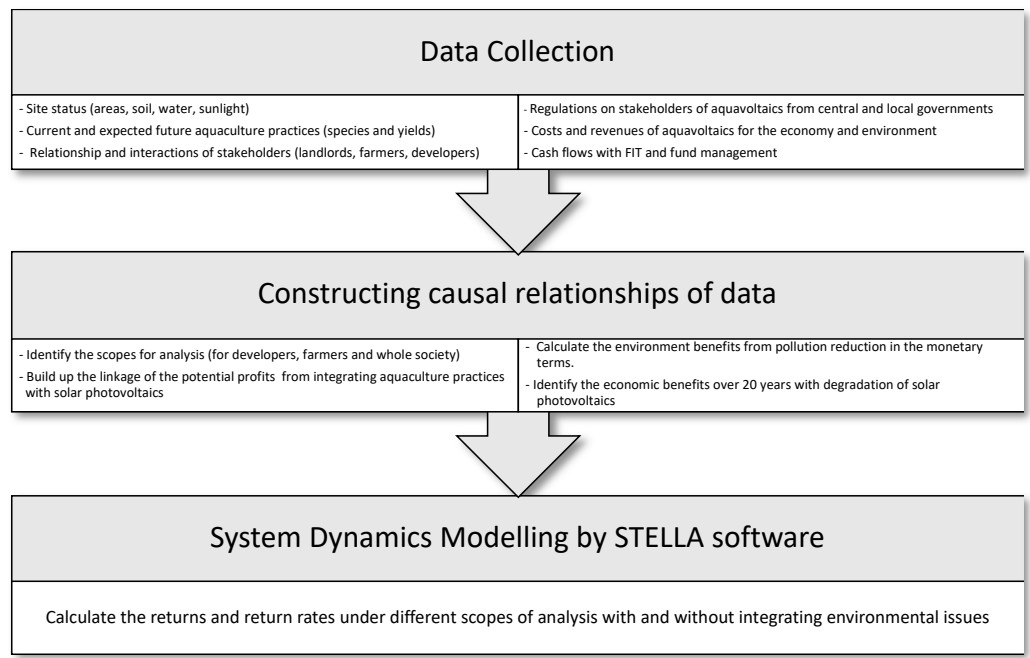

**Figure 1.** Steps of system dynamics modeling for aquaculture photovoltaics.

To calculate the return on investment (IRR) combined with the additional environmental benefits from the symbiosis of aquaculture and solar PV, we employed system dynamics, a computer simulation method developed by Professor Forrester in the United States in 1950 [7]. This approach treats all phenomena as a system, enabling a comprehensive understanding of the dynamic nature of complex phenomena that may not be intuitively evident. System dynamics can be applied to various fields, including business operations, economic development, environmental change, social unrest, urban recession, psychology, and physiology [8]. In this study, we utilized the Stella Architect software v.2.1.2 developed by isee systems (https://www.iseesystems.com/, accessed on 1 March 2021) to build the system dynamics model, representing the parameters, economic factors, and environmental benefits. Three sub-models were constructed within the system dynamics model, following the principles outlined below:

- Cost assessment sub-model

This sub-model estimates the total costs associated with photovoltaic installation during construction, operation, and maintenance, as well as the costs of farm shed installation and fish farming activities.

- Benefit assessment sub-model

The benefit assessment comprises two components: economic benefits and environmental benefits. Economic benefits include income from photovoltaic feed-in-tariffs (FIT) and farming production. Environmental benefits are monetized based on the reduction in carbon emissions and air pollution resulting from the replacement of fossil fuels with solar-generated electricity.

- Net benefit assessment sub-model

The economic costs and benefits are balanced to determine the net economic benefit. This is then combined with the environmental benefit to evaluate the overall net benefits for the environment and economy. The whole scheme of analysis is indicated as Figure 2.

To provide a clearer assessment of the investment benefits of symbiotic practices in terms of both economic and environmental aspects, this study also includes the evaluation

of the return on investment, specifically the internal rate of return (IRR), for the symbiosis scenario. The evaluation formula for IRR is expressed as follows:

$$\sum_{t=0}^{20} \frac{C_t}{(1+i)^t} = 0 \tag{1}$$

where

$i$     IRR, is the discount rate at which the net present value is 0 at the period end ($t = 20$);

$Ct$ is   net cash flow in year $t$, (aquaculture income + FIT income—operation and maintenance cost of farming facilities—operation and maintenance cost of photovoltaic facilities—annual payment of loans for farming facilities and photovoltaic facilities)

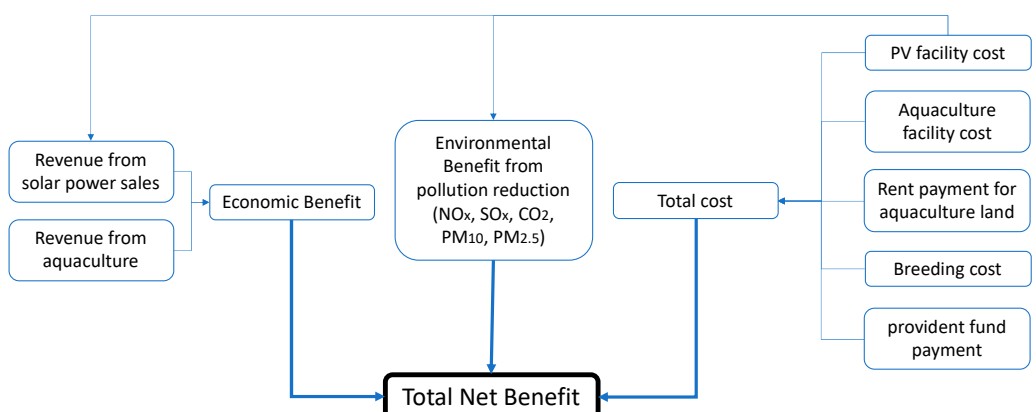

**Figure 2.** Schematic diagram of cost-benefit analysis for aquaculture photovoltaics in SDM.

In addition, we built an environmental benefit module of System Dynamics (Figure 3) and used the following formula for the quantitative evaluation:

$$\text{Environmental benefit (TWD/year)} = \Sigma \text{ (annual pollution emission reduction} \times \text{unit environmental cost)} \tag{2}$$

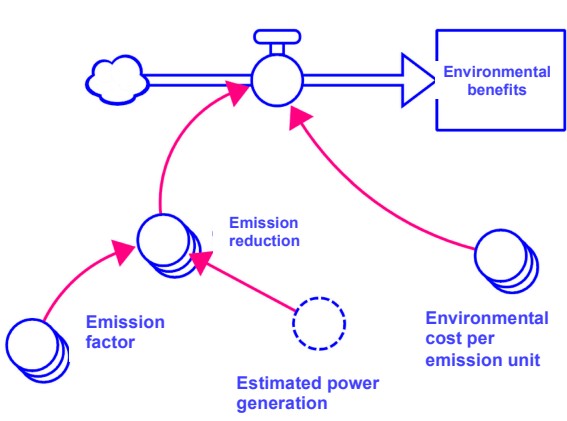

**Figure 3.** Schematic diagram of cost-benefit analysis for aquaculture photovoltaics.

As indicated in Table 1, the aquaculture–electricity symbiosis, with 1 MW of power generation, can effectively reduce annual emissions by 662 tons of $CO_2e$, 0.25 tons of SOx, 0.362 tons of NOx, 0.016 tons of PM2.5, and 0.02 tons of PM10. When translated into monetary values, this practice can contribute to an environmental benefit equivalent of approximately TWD 1.674 million per year (USD 57,000 per year). Over a 20-year period, the total environmental benefits would amount to TWD 33.49 million (USD 1.15 million).

**Table 1.** Estimated environmental benefits from 1 MW of Solar PV power.

| Emission Variety for Reduction | Tons/Year (A) | Pollutant Abatement Cost (TWD/Ton) * (B) | Annual Environmental Benefit from Emission Reduction (TWD Million/Year) ** (A × B) |
|---|---|---|---|
| CO2 | 662 | 1913 | 1.267 |
| SOx | 0.25 | 26,000 | 0.006 |
| NOx | 0.362 | 28,000 | 0.01 |
| PM2.5 | 0.016 | 24,350,000 | 0.39 |
| PM10 | 0.02 | 60,000 | 0.01 |
| Total | - | - | 1.674 |

* Summarized from: [9–15]. ** The total environmental benefit of the total period (20 years) can reach approximately TWD 33.49 million.

## 3. Scenarios and Assessment Results

### 3.1. Scenario Setting

Based on the conditions set for the evaluation of fish farming and solar PV–electricity symbiosis in this study, the following key points are outlined:

1. Total investigated area

This assessment covers 538 hectares of land operated by 55 farmers in the Tainan area, specifically in Qigu District and Beimen District. Out of the total area, 444 hectares are used for aquaculture and 227 hectares for photovoltaic facilities. There are 580 landowners and 55 participating farmers.

2. General status of aquaculture

The predominant form of aquaculture in the area is outdoor farming, including shallow flatbed areas for clam cultivation (68%), deep flatbed areas for tilapia (10%), mullet (8%), white shrimp (7%), milkfish (4%), grouper (2%), and yellow trevally (<1%). The fish species for breeding remain the same as before implementing the aqua-PV practice, but the proportion of white shrimp has been increased for better economic returns. Collaboration with industries and universities in the region is also present to support farmers in upgrading their white shrimp farming through the installation of column-type photovoltaic facilities.

3. Operating scenario for the symbiosis case

This study assumes a ground-based photovoltaic installation capacity of 1 MW per hectare, resulting in a total installation capacity of 227 MW. As the area is mainly used for outdoor farming, column-type photovoltaics are employed for a 20-year operation period.

4. Business model of the symbiotic practice

The major stakeholders in this case study include landlords, farmers, and photovoltaic system operators. Landlords lease their land to photovoltaic system operators and collect rent. The farmers, who were previously engaged in aquaculture without symbiosis, now pay rent to the photovoltaic system operators. The rent is directed towards the "Aquaculture Development and Management Fund", managed by a committee responsible for determining fund allocation. Farmers receive 100% of the farming income.

Photovoltaic system operators must assess the suitability of the aquaculture–electricity symbiosis based on the physical characteristics of the aqua-PV cases (area, topography, electricity feeder, aquacultural resources, ecology, etc.). According to the law, "Social Inspections" are required for cases with a capacity of over 2 MW to ensure compliance with the procedure and obtain permission from the landlords. The original farmers also need written consent to participate in the aquaculture-photovoltaics plans. They then collaborate with the photovoltaic system operators to jointly plan and establish photovoltaic facilities, which allows them to apply for feed-in-tariffs (FIT). The "Applying for Agricultural Land as Agricultural

Facilities Allowable Use Review Measures" [16] outlines the requirements for facility area ratio, shade ratio, and other local regulations that must be met for aquaculture facilities.

Based on the symbiosis model, when dealing with a large area of land involving multiple landlords, photovoltaic system operators step in to lease the land from the landlords at a more favorable price. Additionally, the operators pay rent to participate in the planning and construction of the project site. During the maintenance period, there are cash flows (loans) to assist farmers in complying with regulatory requirements for fish harvests and ensuring the stability of photovoltaic power generation. The main source of income for the photovoltaic system operators is the FIT.

5. Cost-benefit Analysis:

This study conducts cost-benefit evaluations from the perspectives of photovoltaic system operators, participating farmers, and society as a whole. The analysis considers various factors such as loans and subsidies. Detailed information can be found in Table 2, which summarizes the cost-benefit evaluations.

**Table 2.** Background setting for evaluation.

| Perspective | Whole of Society | Photovoltaic System Operators | Farmers |
|---|---|---|---|
| Business model [a] | 1. Photovoltaic system operators pay rent to landlords and receive usage funds from farmers. 2. The purposes of the usage funds are determined by a committee composed of farmers. 3. Participating farmers are the original farmers who need to upgrade their aquaculture facilities. 4. Outdoor farming is the major type of practice with vertical ground PV facilities. | | |
| Case area [a] (hectares) | 538 | | |
| Fish school area [a] (hectares) | 444 | | |
| No. of participating farmers [a] | 55 | | |
| Aquaculture types | Shallow bed (Clam) and deep bed (Milkfish, Snapper, Grouper) are both cultivated. In addition, the local industry–university cooperation model assists farmers in the indoor cultivation of white shrimp. | | |
| Solar PV facilities area [a] (hectares) | 227 | | |
| PV capacity [b] (MW) | 227 | | |
| Duration (year) | 20 | | |
| Loan proportion (%) for aquaculture | - | - | 80 |
| Loan interest rate [c] (%) for aquaculture | - | - | 1.235 |
| Subsidy for aquaculture [d] (TWD 1000) | - | - | 500 |
| Loan proportion (%) for solar PV | - | 80 | - |
| Loan interest rate [e] (%) for solar PV | - | 3.42 | - |

[a]. Refers to the 5 cases of aquaculture photovoltaics plans approved by the Council for Agriculture in April and June 2019 in Qigu District and Beimen District in Tainan. [b]. An estimate of 1 hectare land area is required for 1 MW of ground photovoltaics. [c]. The annual loan interest rate is referred to in Article 6 and Schedule 2 of Article 24 of the "Measures for Handling Policy-Based Agricultural Project Loans (21 October 2020 Amendment)" [16]. The annual interest rate is reduced by 0.055%, and the interest rate of the project announced by the Agricultural Finance Bureau is 1.29% [17]. So the interest rate is set as (1.29–0.055) % = 1.235%. [d]. The Council for Agriculture has set up a one-time subsidy for facilities related to symbiosis. If the maximum subsidy for each outdoor farming household is TWD 500,000 [18], with 55 farmers in total, the amount of the subsidy will be around TWD 27.5 million. [e]. For the annual interest rate on photovoltaic facilities loans, the current market rate for the total annual cost of solar photovoltaic facilities financing is about 3.42%.

### 3.2. Assessment Results

The aquaculture photovoltaic projects with a total capacity of 227 MW generate significant environmental benefits. These projects contribute to an annual reduction of 150,393 tons of $CO_2e$ emissions, 56.8 tons of SOx emissions, 82.3 tons of NOx emissions, 3.7 tons of PM2.5 pollution, and 4.6 tons of PM10 pollution. In monetary terms, the total annual environmental benefits amount to TWD 381.322 million, or TWD 7.63 billion over a 20-year period.

From an investment standpoint, the system dynamics model, illustrated in Figures 2 and 3, is used to evaluate the cost-benefit of the 227 MW aquaculture photovoltaic projects. As shown in Table 3, over the entire 20-year period, society as a whole benefits the most in terms of net environmental–economic benefits from the symbiotic approach (TWD 5726 million). This is followed by the benefits from the feed-in-tariffs (FIT) policy investment (TWD 3737 million), benefits to the photovoltaic system operators (TWD 2696 million), and finally, benefits to the participating farmers (TWD 644.39 million).

**Table 3.** Cost-benefit and IRR from the symbiosis of aquaculture and 227 MW of solar power.

| | Item | For the Whole of Society | For Photovoltaic Operators | For the FIT Policy | For Farmers |
|---|---|---|---|---|---|
| **Cost** | Total cost per unit of aquaculture facility upgrade ab (TWD 1000/hectare) | 8500 | - | - | 8500 |
| | Annual payment for aquaculture loan (TWD 1000/year) | - | - | - | 19,410 |
| | The total cost of an aquaculture unit (TWD 1000/hectare-year) | 374.6 | - | - | 374.6 |
| | Rent (TWD 1000/hectare-year) | 400 | 400 | - | (included in the total cost) |
| | Unit cost of PV facilities (TWD 1000/kW) (A) | 46.8 | 46.8 | 41.8 | - |
| | Annual cost for PV facility setup loan (TWD 1000/year) | - | 587,300 | 587,300 | - |
| | PV facility operation and maintenance cost percentage (% of A) | 1 | 1 | 3.62 | - |
| **Economic benefits** | Aquaculture income (TWD 1000/hectare-year) | 444.6 | - | - | 444.6 |
| | Aquaculture Development and Management Fund (TWD 1000/year) | 5770 | - | - | 5770 |
| | FIT rate (TWD/kWh) | 3.9849 | 3.9849 | 3.9849 | - |
| **20-year economic benefits (TWD 1000) (B)** | | 26,562,510 | 22,729,710 | 22,729,710 | 4,063,600 |
| **20-year economic cost (TWD 1000) (C)** | | 20,835,970 | 20,414,780 | 19,373,900 | 3,800,530 |
| **20-year economic net benefits (TWD 1000/year) (B-C)** | | 5,726,540 | 2,314,930 | 3,355,810 | 263,070 |
| **Return Rate on Investment (IRR) (20 -year period)** | | 4.34% | 8.34% | 12.54% | 19.75% |

As shown in Table 4, the return on investment in terms of net economic benefit for the aquaculture photovoltaic projects, from the perspective of society as a whole, is the highest at approximately 4.34%. When considering the integrated environmental benefits,

the environmental-economic return on investment (IRRe) increases to 9.09%. This means that, for society as a whole, the aquaculture–electricity symbiosis approach is not only economically viable but also environmentally sustainable. While farmers may appear to have the lowest net economic benefits due to the smaller scale of their cost-benefit, they have the highest return on investment among all scenarios, reaching 19.75%. The second-highest IRR is for the photovoltaic system operators. From the perspective of the FIT policy in Taiwan, the IRR of aquaculture–electricity symbiosis can reach 12.54%. However, for the photovoltaic system operators, the IRR drops to 8.34%. Nevertheless, both farmers and photovoltaic industry operators achieve higher IRR than the IRR for society as a whole, demonstrating that adopting aquaculture photovoltaics is economically viable for both parties.

**Table 4.** Environmental benefits and synthetic IRR from the symbiosis of aquaculture and 227 MW of solar power.

| Item | | For the Whole of Society (Baseline Scenario) | For the Photovoltaics Industry | For the FIT Policy | For Farmers |
|---|---|---|---|---|---|
| **Environmental Benefit Items** | Carbon reduction | 150,393.6 tons $CO_2$e/ year $\xrightarrow{Monetization}$ TWD 287.853 million/year | | | |
| | SOx reduction | 56.8 tons/year $\xrightarrow{Monetization}$ TWD 1.462 million/year | | | |
| | NOx reduction | 82.3 tons/year $\xrightarrow{Monetization}$ TWD 2.317 million/year | | | |
| | PM2.5 reduction | 3.7 tons/year $\xrightarrow{Monetization}$ TWD 89.415 million/year | | | |
| | PM10 reduction | 4.6 tons/year $\xrightarrow{Monetization}$ TWD 274,000/year | | | |
| **20-year economic net benefit (TWD 1000) (A)** | | 5,726,540 | 2,314,930 | 3,355,810 | 263,070 |
| **20-year relative environmental benefits (TWD 1000) (B)** | | 381,322 | 381,322 | 381,322 | 381,322 |
| **20-year environmental—economic net benefit (TWD 1000) (A + B)** | | 6,107,862 | 2,696,252 | 3,737,132 | 644,392 |
| **Return Rate on Investment (IRR) (20-year period)** | | 4.34% | 8.34% | 12.54% | 19.75% |
| **Environmental—Economic Return Rate on Investment (IRRe) (20-year period)** | | 9.09% | - | - | - |

Data source: summarized from the modeling results.

## 4. Conclusions

The results obtained from the system dynamics modeling indicate that, over the 20-year period, the symbiotic relationship between rural power generation and aquaculture has resulted in environmental–economic net benefits of TWD 5726 million (approximately USD 191 million) for society as a whole. Photovoltaic system operators have gained TWD 2696 million (approximately USD 90 million), while participating farmers have benefited with TWD 644 million. The net economic return on the symbiosis of aquaculture and photovoltaics, from the perspective of society as a whole, is approximately 4.34%. By considering the integrated environmental benefits, the environmental-economic return on investment (IRRe) can be increased to 9.09%. This demonstrates that aquaculture–electricity symbiosis is not only economically viable but also environmentally beneficial for society as a whole. Despite farmers appearing to have the lowest net economic benefit, they have the highest IRR of 19.75% among all stakeholders. The photovoltaics industry achieves an IRR of 8.34%.

Furthermore, this study reveals that the symbiotic relationship between aquaculture and electricity can serve as an effective tool for adaptation in the face of a changing climate. However, there are certain risk issues that remain challenging to address within the current policy framework, which limit the scope of this cost-benefit assessment study. These issues include:

1.  The aquaculture photovoltaics symbiosis business model is more complex compared to standalone photovoltaic systems.
2.  Rent for roofs or land is susceptible to price speculation, posing a risk of escalating costs for symbiotic projects.
3.  Mutual trust between aquaculture farmers and photovoltaic system operators needs improvement through alternative mechanisms.
4.  This assessment does not reflect the actual costs of symbiotic practices due to the reduction of the feed-in-tariffs. Additionally, the government should proactively promote and encourage symbiosis between the aquaculture and electricity industries.

While this study proposes a feasible analysis framework for evaluating the costs and benefits of aquavoltaics, it acknowledges the complexity of social and financial aspects associated with these practices. Further refinement of the modeling results can be achieved by considering additional factors. For instance, the high humidity conditions under aquavoltaics operation can reduce the lifespan of photovoltaic modules. Additionally, advancements in photovoltaic technology, such as longer lifespans and higher power output rates, can improve the economic and environmental efficiency of the system.

**Author Contributions:** Conceptualization, L.W.; methodology and software, C.-H.L. and Y.-C.L.; validation, C.-H.L.; formal analysis, L.W.; investigation, Y.-C.L.; data curation, Y.-C.L.; writing— original draft preparation, Y.-C.L. and C.-H.L.; writing—review and editing, C.-H.L.; visualization, C.-H.L.; supervision, L.W.; project administration, L.W. All authors have read and agreed to the published version of the manuscript.

**Funding:** This research received no external funding.

**Institutional Review Board Statement:** Not applicable.

**Informed Consent Statement:** Not applicable.

**Data Availability Statement:** No new data were created.

**Conflicts of Interest:** The authors declare no conflict of interest.

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
