# Peer review of "Are Aquavoltaics Investable? A Framework for Economic and Environmental Cost-Benefit Analysis"

_sustainability, doi:10.3390/su15118965_

Round 1
Reviewer 1 Report
Communication presented by the authors is an economic calculation of the feasibility of using aquavoltaics in Taiwan. The topic of Communication is relevant and may be of interest to specialists and researchers in the fields of agriculture, economics, sustainable development and solar energy. Communication provides an interesting model, as well as calculations based on it, however, as comments and recommendations, several points should be noted:
1. To enhance the work and improve its visual perception, the authors should add a scheme and principle of operation in general terms of the system under consideration.
2. What is the rationale behind the authors' choice of Stella Architect software for research?
3. Under aquavoltaics operating conditions, there is high humidity, which can reduce the service life of photovoltaic modules, most of which have a rated power life of 15-20 years - was accelerated degradation taken into account in the model proposed by the authors? At the same time, today, photovoltaic modules with a rated power of 40-50 years already exist and are commercially produced (for example, 10.4018/IJEOE.2020040106 , etc.), which can improve the economic efficiency of the system under consideration according to the model proposed by the authors - these two points should be briefly described by the authors in the paper.
4. There is no reference to Figure 1 in the text of the work. The text in Figure 1 should be enlarged. In Table 2, the "*" character is present, but its meaning is not described. Table 4 is not referenced in the text.
5. Do the authors plan to apply and implement the results obtained in the work somewhere and somehow? In what direction are their further work and research planned?
In general, presented Communication leaves a positive impression, however, it is not without minor shortcomings. After eliminating these comments and taking into account the recommendations made, the Communication presented may be of interest to readers of the journal "Sustainability".
The text of the authors' work should be proofread by a native English speaker.
Author Response
Many thanks for the comments. Responses from authors are as follows.
- To enhance the work and improve its visual perception, the authors should add a scheme and principle of operation in general terms of the system under consideration.
Many thanks for the suggestion. A scheme for this project has been added in the revision draft for better understanding. Hopefully this visual representation will significantly enhance the clarity and understanding of the project for readers.
- What is the rationale behind the authors' choice of Stella Architect software for research?
Stella Architect is a system dynamics modeling software developed by isee systems. The software incorporates the principles of system dynamics, such as stocks, flows, and feedback loops, into the modeling process. By constructing system dynamics models in Stella Architect, users can simulate and experiment with different scenarios to understand the behavior of the system under various conditions. This allows for a deeper understanding of the underlying dynamics and helps in identifying leverage points for intervention or policy changes to improve system performance.
- Under aquavoltaics operating conditions, there is high humidity, which can reduce the service life of photovoltaic modules, most of which have a rated power life of 15-20 years - was accelerated degradation taken into account in the model proposed by the authors? At the same time, today, photovoltaic modules with a rated power of 40-50 years already exist and are commercially produced (for example, 10.4018/IJEOE.2020040106 , etc.), which can improve the economic efficiency of the system under consideration according to the model proposed by the authors - these two points should be briefly described by the authors in the paper.
The authors express their gratitude to the reviewer for their valuable suggestion, which has provided insightful input. The suggestion specifically addresses the aspects of humidity and module lifetime, and the authors acknowledge the potential value of incorporating these considerations into the model in future iterations. These points for further improvement in model construction have been duly noted and emphasized at the conclusion of the revised draft.
- There is no reference to Figure 1 in the text of the work. The text in Figure 1 should be enlarged. In Table 2, the "*" character is present, but its meaning is not described. Table 4 is not referenced in the text.
Fig 1 in the first draft illustrating the study process of this study, not referring from other references, is now Fig 2 with revision in the new version for better understanding. Table 4 is the summary of the modeling results in this study, also not from other references.
- Do the authors plan to apply and implement the results obtained in the work somewhere and somehow? In what direction are their further work and research planned?
Given the immense pressure arising from the net-zero emission policy, there is a need to identify and evaluate potential and feasible approaches within society. The presented paper showcases a cost-benefit analysis framework that integrates economic and environmental factors. This framework has the potential to be applied to a wide range of policy or program evaluations. However, it is important to note that for each specific case, there may be numerous interconnections among system elements that require careful consideration. These additional linkages should be thoroughly examined to ensure a comprehensive evaluation of the given case.
In general, presented Communication leaves a positive impression, however, it is not without minor shortcomings. After eliminating these comments and taking into account the recommendations made, the Communication presented may be of interest to readers of the journal "Sustainability".
The text of the authors' work should be proofread by a native English speaker.
Thanks for the positive opinions and advice for our paper. The paper has again been carefully proofread and some minor rephrasing has been done in the new version to ensure its quality and clarity.
Reviewer 2 Report
The study seems that it might result with potentially very interesting knowledge. However, the paper itself has a number of few flaws, which need to be corrected, and the paper has to be improved before it can be accepted for the publication.
It is very important to investigate the environmental and biological, in addition to the economics. However, it seems most of the results are demonstrated by the previous literatures, and the body of the work need to be written in a more relevant way.
The major concerns need to be addressed:
1. If it were intended to be research communication article, I would have expected more description and discussion of the results. Only the most relevant literature should be discussed in the introduction and discussion. Data analysis is not complete, same for figures and tables.
2. The methodology is not clearly described. The methods are not sufficiently detailed or precise. The test criterion is whether a researcher who wishes to repeat the work could do so without introducing other variability. The economic analysis is not clear and, since the strength of the conclusions depends on them, this is an essential requirement.
3. Results. I could not reconcile the presentation of Methods and the Tables and Figures presenting the data and their analysis. It could be that the text does not describe it well but there could also be an error in the reporting. This needs to be checked and clearly presented. It will be easier to provide a critique on this section if the methods and economic analysis are clarified.
4. In relation to the discussion, some passages are purely speculative, whose hypotheses are weakly supported. The Discussion covers some really interesting points but it needs a clearer direction as to why they are of significance and interest to the main thrust of the paper. Since this is not elaborated (see above), their inclusion appears somewhat random and disorganized.
5. The conclusion of the manuscript is superficial and purely descriptive. This could relate to the original hypotheses, what the tests show and what the implications are.
Author Response
The study seems that it might result with potentially very interesting knowledge. However, the paper itself has a number of few flaws, which need to be corrected, and the paper has to be improved before it can be accepted for the publication.
It is very important to investigate the environmental and biological, in addition to the economics. However, it seems most of the results are demonstrated by the previous literatures, and the body of the work need to be written in a more relevant way.
Many thanks for the suggestions and opinions regarding the overall quality of this paper. The paper has been thoroughly revised, taking into careful consideration the reviewer's five specific comments, particularly in the introduction and discussion sections. Furthermore, a scheme depicting the project has been included in the revised draft to facilitate better comprehension. It is anticipated that this new version will greatly improve the clarity and understanding of the project for readers.
Reviewer 3 Report
A graphic or photo image of the PV's on columns in the ponds would have been very useful. The authors mention that the practical impacts on aquaculture operations need to be documented and farmer trust with the PV operators needs to be improved. A bit more discussion of this point would be instructive.
Overall, very well written paper with excellent English. Original work which should be published.
Author Response
A graphic or photo image of the PV's on columns in the ponds would have been very useful. The authors mention that the practical impacts on aquaculture operations need to be documented and farmer trust with the PV operators needs to be improved. A bit more discussion of this point would be instructive.
Overall, very well written paper with excellent English. Original work which should be published.
Many thanks for the review’s suggestions and encouraging words. The paper has undergone a thorough revision, with special attention given to the reviewer's comments, particularly in the introduction and discussion sections. Due to high social complexity of the aquavoltaics practices and given the original purposes of this study, to simply the principal idea of this paper, it is in fact a pity that quite a few details are not further addressed in this paper or included in this study, which can be further improved in the future.
Round 2
Reviewer 2 Report
The authors improved the paper, accepted my suggestions, in view of this, I consider it approved for publication.